# GSEYOLOX-s: An Improved Lightweight Network for Identifying the Severity of Wheat Fusarium Head Blight

Rui Mao [1], Zhengchao Wang [1], Feilong Li [1], Jia Zhou [2], Yinbing Chen [1] and Xiaoping Hu [2,3,*]

1 College of Information Engineering, Northwest A&F University, Yangling, Xianyang 712100, China
2 College of Plant Protection, Northwest A&F University, Yangling, Xianyang 712100, China
3 Key Laboratory of Integrated Pest Management on Crops in Northwestern Loess Plateau, Ministry of Agriculture and Rural Affairs, Yangling, Xianyang 712100, China
* Correspondence: xphu@nwafu.edu.cn

**Abstract:** *Fusarium* head blight (FHB) is one of the most detrimental wheat diseases. The accurate identification of FHB severity is significant to the sustainable management of FHB and the guarantee of food production and security. A total of 2752 images with five infection levels were collected to establish an FHB severity grading dataset (FHBSGD), and a novel lightweight GSEYOLOX-s was proposed to automatically recognize the severity of FHB. The simple, parameter-free attention module (SimAM) was fused into the CSPDarknet feature extraction network to obtain more representative disease features while avoiding additional parameters. Meanwhile, the ghost convolution of the model head (G-head) was designed to achieve lightweight and speed improvements. Furthermore, the efficient intersection over union (EIoU) loss was employed to accelerate the convergence speed and improve positioning precision. The results indicate that the GSEYOLOX-s model with only 8.06 MB parameters achieved a mean average precision (mAP) of 99.23% and a detection speed of 47 frames per second (FPS), which is the best performance compared with other lightweight models, such as EfficientDet, Mobilenet-YOLOV4, YOLOV7, YOLOX series. The proposed GSEYOLOX-s was deployed on mobile terminals to assist farmers in the real-time identification of the severity of FHB and facilitate the precise management of crop diseases.

**Keywords:** *Fusarium* head blight; wheat diseases severity identification; lightweight network architecture; YOLOX; attention mechanism

## 1. Introduction

Wheat (*Triticum aestivum* L.) is one of the world's three major food crops. In 2021, global wheat production exceeded 776 million tons, of which China, with 136 million tons ranked first, accounting for ca. 17.6% of the world's total wheat production [1]. However, *Fusarium* head blight (FHB) is a destructive disease that restricts the safety of wheat production and food quality [2,3]. In epidemic years, FHB can reduce the yield loss of wheat by 10–70%, influencing more than 7 million hectares of wheat-planting areas [4]. The classification of FHB severity is crucial in making decisions on its control. The inaccurate identification of FHB severity will lead to ineffective control and fungicide abuse [5]. Therefore, it is essential to conduct a precise method to identify the severity of FHB.

Traditionally, the identification of the severity of FHB mostly depends on technicians' continuous field investigation. Disease severity based on scale proportions is usually estimated using the naked eye depending on the number or the area of the lesions in the whole wheat ear. This method is not only time-consuming but also biased and unreliable. Deep convolutional neural networks (DCNNs) [6] are state-of-art neural networks, which have the ability of self-learning and contribute to the automatic recognition and severity estimation of crop diseases. Esgario et al. [7] used the ResNet50 model with 25 MB parameters to estimate the severity caused by four diseases on coffee leaves, such as

cercospora leaf spot, brown leaf spot, rust, and leaf miner, which achieved an 86.51% accuracy. However, the average time required by this model to train one epoch was 21.90 s. Ji et al. [8] employed images of single crop leaves with a simple background to identify ten crop disease types and three crop disease severity levels (normal, general, and serious samples), and adopted the ResNet50 model improved by multi-label binary correlation CNN (BR-CNN) with an accuracy of 86.70% and the parameters of 23 MB. Mi et al. [9] developed a novel GradCAM++ network based on a densely connected convolutional network (DenseNet) and a convolutional block attention module (CBAM) to identify the severity of wheat stripe rust and the accuracy of the severity level improved to 97.99%. Zhang et al. [10] proposed a segmentation model of single wheat ear and the FHB lesion segmentation model based on their self-constructed IABC-K-PCNN. According to the radio between the lesion and the whole wheat ear, the speed of the FHB classification was about 5 s per photo, with an accuracy of 92.5%. Although satisfactory results are reported in the above studies based on large CNNs, there will be a problem in the practical application of disease severity identification due to the high number of parameters, the large storage space, and computational consumption.

In order to reduce the parameters and speed up network training, some lightweight CNNs have been gradually applied in plant disease detection research. Bao et al. [11] developed SimpleNet using convolution and inverted residual blocks that achieved a 94.10% recognition accuracy of wheat ear diseases. Hong et al. [12] improved the YOLOv4 by a lightweight network, non-maximum suppression (NMS), and complete intersection over union (CIoU) to achieve model lightweight improvement and deploy to UAV to identify FHB. Liu et al. [13] improved the YOLOX-Nano model by introducing blueprint-separable convolution (BSConv), attention mechanism, and the asymmetric ShuffleBlock. This study provided a feasible solution for the real-time detection of apple leaf disease with a 91.08% accuracy. Although the parameters of the above network models were smaller than those of the common DCNNs, these models did not directly identify the severity of FHB. In recent years, researchers have started to concentrate on extracting critical feature information and improving the performances of models by incorporating the attention mechanism into the networks. Cui et al. [14] introduced the CBAM into the autoencoder to identify maize leaf diseases from PlantVillage in laboratory scenarios, and obtained a 99.44% identification accuracy. Li et al. [15] applied a hybrid attention, the Atrous Space Pyramid Pool (ASPP), to optimize DeepLab V3+ to accurately segment lesions and automatically assess the severity of cucumber downy mildew and powdery mildew. The above studies demonstrated that using different strategies to optimize neural networks can further improve the recognition accuracy and training speed. However, considering the small and subtle differences between the different severity levels of FHB, it is still a great challenge to establish a real-time and accurate identification model for the severity of FHB on the mobile terminal.

To reduce the parameters while improving the detection accuracy and speed of the classical YOLOX-s, a lightweight network named GSEYOLOX-s was proposed in this study. An FHB severity grading dataset (FHBSGD) was established, which contained a total of 2752 images with five infection levels of FHB under experimental and complex field conditions. GSEYOLOX-s model was designed by fusing a simple, parameter-free attention module (SimAM), the ghost convolution of the model head (G-head), and the efficient intersection over union (EIoU) loss function into YOLOX-s. The three improvements maximized the reduction in model parameters and facilitated the deployment of our model on the mobile terminal. The experimental results show that the proposed GSEYOLOX-s model only had 8.06 MB parameters while achieving a mean average precision (mAP) of 99.23% on the FHBSGD. The proposed model provides support for precise planting and intelligent decision-making for wheat production.

## 2. Materials and Methods

### 2.1. Data Acquisition

FHB consists in the premature fading or wilting of wheat ears that can be seen 3 weeks post *Fusarium* spp. [16]. infection, with infected spikelets showing orange or pink colors due to pathogen spores and mycelia on wheat ears [17]. According to the National Standard of the People's Republic of China for FHB (GB/T 15796-2011), the severity of FHB is divided into 5 levels ranging from Level_0 to Level_4. Level_0 represents disease-free; Level_1 indicates that the number of infected spikelets is less than 25% of all spikelets; Level_2 indicates that the number of infected spikelets is less than 50% and more than 25% of all spikelets; Level_3 indicates that the number of infected spikelets is less than 75% and more than 50% of all spikelets; and Level_4 indicates that the number of infected spikelets is more than 75% of all spikelets.

The images acquisition area is Guanzhong district, Shaanxi Province, such as Wugong, Fuping, Pucheng, and Mei County. The image collection time was from 10 May 2022 to 20 May 2022 during late grouting stage of wheat. The shooting equipment include one digital camera (Canon EOS 850D) and mobile phones (Apple 11pro and Huawei Mate20), using automatic white balance and optical focusing. These devices were held 25–50 cm away from wheat ears and the time of acquiring image was from 9:00 a.m. to 18:00 p.m. The images were stored in JPEG format with a resolution of 6000 × 4000 pixels and 4032 × 3024 pixels. To meet the training requirements of the proposed detection model, the image size was uniformly adjusted to the size of 640 × 640 pixels. According to the above standards, we classified the collected dataset into five severity levels of FHB. The dataset contained a total of 2752 images of FHB, including 556 images of Level_0, 638 images of Level_1, 732 images of Level_2, 598 images of Level_3, and 228 images of Level_4. Specific samples of the FHBSGD are shown in Figure 1.

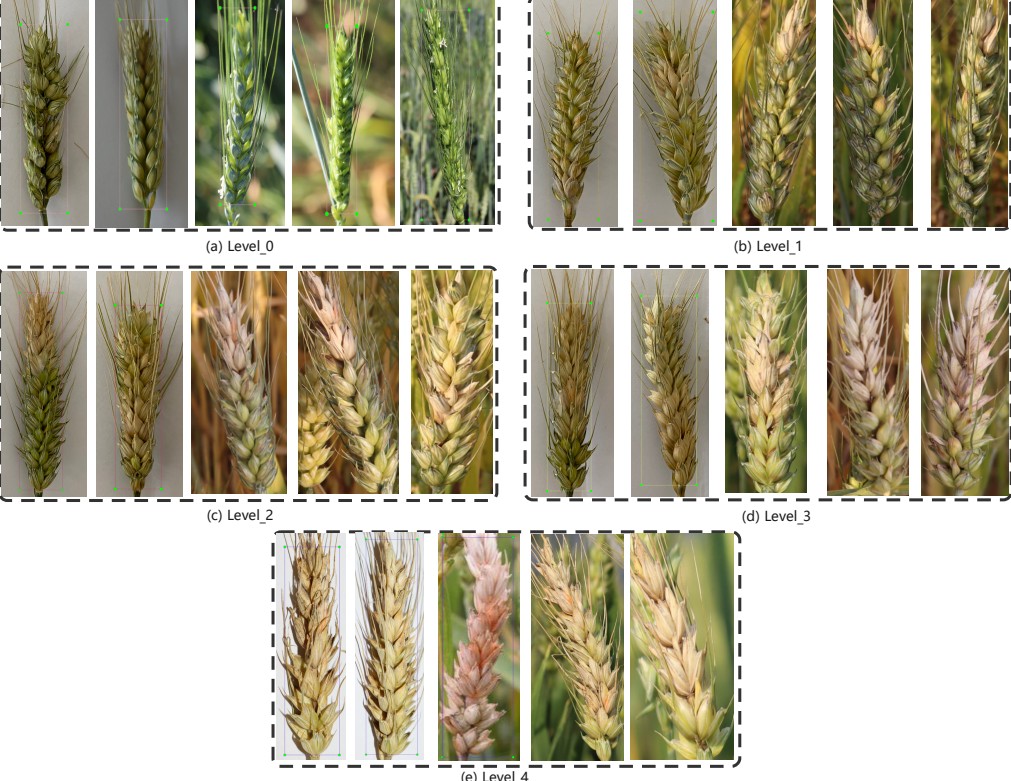

**Figure 1.** Wheat FHB with different severity levels under experimental and field conditions in the dataset.

*2.2. Data Processing*

Dataset annotation is essential for an identification task based on deep learning. Under the guidance of plant protection experts, we used "LabelImg" to manually label the severity grading of wheat ears in the images and generated XML-type annotation files with Pascal VOC 2007 dataset format standard. The dataset was divided into training set, validation set, and test set at 8:1:1, as shown in Table 1. To enrich the background of samples and improve the robustness against image noises, the data enhancement method in this study adopted the combination of Mixup and Mosaic [18], which was helpful to improve the detection ability of partial occlusions and subtle targets.

**Table 1.** The sample distribution in FHBSGD.

|  | **Level_0** | **Level_1** | **Level_2** | **Level_3** | **Level_4** |
|---|---|---|---|---|---|
| training set | 444 | 510 | 586 | 478 | 182 |
| validation set | 56 | 64 | 73 | 60 | 23 |
| test set | 56 | 64 | 73 | 60 | 23 |
| total | 556 | 638 | 732 | 598 | 228 |

*2.3. GSEYOLOX-s for FHB Severity Identification*

2.3.1. The Network Architecture

YOLOX is a new high-performance target-detection model presented by Ge and Liu et al. [19]. Its network architecture baseline includes CSPDarknet [20], feature pyramid networks (FPNs) [21] structure combined with the path aggregation networks (PANs) [22] structure to extract features, the decoupled head with the anchor-free method, and IoU loss [23] for bounding box regression. YOLOX-s, a lightweight version of YOLOX series, was selected as the base network architecture in this study. Although the parameter number of YOLOX-s had been significantly reduced compared to other versions of the YOLOX series, the amount of calculation for the mobile terminal was still greater. Moreover, there is still a challenge to accurately detect the subtle differences between different severity levels of FHB.

A lightweight network based on improved YOLOX-s, named GSEYOLOX-s (Figure 2), was proposed in this study. To fully minimize the parameters while enhancing the detection accuracy and speed of the classical YOLOX-s, three main strategies were designed in our model. Firstly, the attention mechanism, SimAM, was introduced into the CSPDarknet backbone network of YOLOX-s (Figure 2c) to focus on essential features without additional parameters, which can effectively improve the expressive ability of feature extraction by optimizing an energy function to calculate the contribution of each neuron. Secondly, G-head (Figure 2e) was applied to take full advantage of redundant feature information among similar images, thus greatly reducing parameters and ensuring the performance of the model. In addition, EIoU function was employed for rapid convergence and more accurate localization of wheat ear lesions.

2.3.2. A Simple, Parameter-Free Attention Module (SimAM)

Referring to the attention module of the human brain, Yang and Zhang presented a novel SimAM [24] based on the theory of neuroscience, which can directly calculate the full three-dimensional weights on the feature graph. The SimAM improved the performance of convolutional neural networks without referring to any additional parameters to calculate the attention weight on the feature map. In this study, we introduced the SimAM to the three feature maps extracted by CSPDarknet. This operator was used to boost the representation ability of feature extraction and a favorable effect for FPN and PAN structure feature fusion. The calculation procedure of the SimAM is shown in Figure 2g.

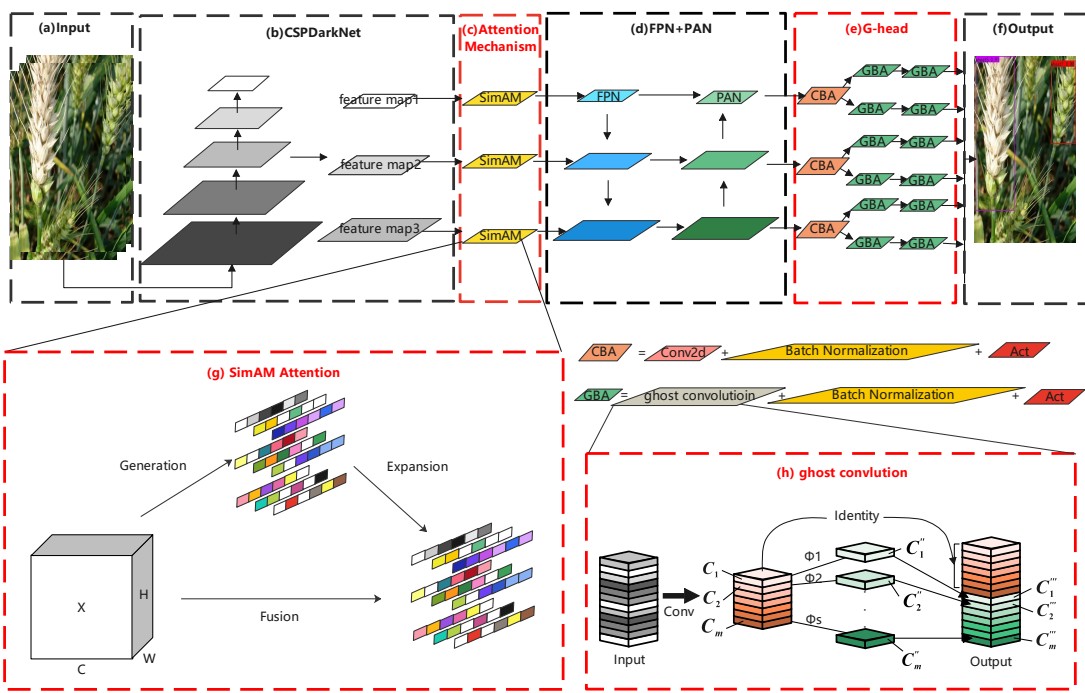

**Figure 2.** The structure of the GSEYOLO-s.

In the theory of neuroscience, the attention of the human brain evaluates the importance of each neuron, in which info-rich neurons usually display different firing modes from surrounding neurons. Furthermore, the activated neurons usually inhibit the surrounding neurons, which need to be given more attention. The study defines the energy of neurons by measuring the linear separability between neurons, which is simplified to produce the following energy function (Equation (1)) [24] to assess the importance of each neuron.

$$e_t^* = \frac{4(\hat{\sigma}^2 + \lambda)}{(t - \hat{u})^2 + 2\hat{\sigma}^2 + 2\lambda} \tag{1}$$

where $\hat{u} = \frac{1}{M}\sum_{i=1}^{M} x_i$, $\hat{\sigma}_t^2 = \frac{1}{M}\sum_{i=1}^{M}(x_i - u_t)^2$, $x_i$ is the target neuron, and $M$ is the number of neurons on this channel. The significance of neurons can be gained by $1/e_t^*$, where $e_t^*$ represents the energy value of neurons at the current position. Combined with the neuroscience theory, the lower the current neuronal energy, the greater the discrepancy between the current neuron and the surrounding neuron, the greater the importance. Then, a scaling operator refines the whole feature.

$$\widetilde{X} = sig\left(\frac{1}{E}\right) \odot X \tag{2}$$

In Equation (2), $sig$ is the sigmoid activation function, $E$ is the integration of all $e_t^*$ across spatial and channel dimensions, and $X \in R^{C \times H \times W}$ is the input features. The sigmoid activation function is applied to limit the E-value range without affecting the relative importance of each neuron.

### 2.3.3. G-head Module

Traditional convolution processing has the redundancy problem of feature maps, resulting in a large amount of computational costs. The ghost convolution designed simple operations to simplify the redundant calculation process of traditional convolution [25]. By introducing the ghost convolution into the G-head, we achieved the lightweight and speed improvement of the model.

The ghost convolution (Figure 2h) first uses a primary convolution to generate $m$ intrinsic feature maps according to the input feature information where $m$ is less than the desired n feature maps; then, each intrinsic feature is subjected to a series of cheap linear operations to generate $s$ ghost features according to the following function:

$$C_i''' = \Phi_i\left(C_j''\right) \qquad \forall i = 1, \ldots, s, \ j = 1, \ldots, m \tag{3}$$

where $C_j''$ is the $j$-th intrinsic feature map generated by primary convolution, $\Phi_i$ is the $i$-th linear operation that generates the $i$-th ghost feature map $C_i'''$, and $C_j''$ generates total $s$ corresponding ghost feature maps through different mappings. Finally, these feature maps are linked together to form the desired n feature maps, which reduces the learning cost of non-critical features without reducing the accuracy of the model.

Equations (4) and (5) represent the calculation of parameter numbers of ordinary convolution $(P_{bsc})$ and ghost convolution $(P_{ghc})$, respectively.

$$P_{bsc} = c_1 c_2 k \times k \tag{4}$$

$$P_{ghc} = \frac{1}{2}\,c_1 c_2 k \times k + \frac{1}{2} c_2 k \times k = \frac{1}{2} c_1 c_2 k \times k \tag{5}$$

where $c_1$ is the channel of the input feature map, $c_2$ is the channel of the output feature map, and $k \times k$ is the size of the convolution kernel. The ghost convolution model has only half as many parameters as the conventional convolution for the same input–output feature maps.

### 2.3.4. Efficient Intersection over Union (EIoU) Loss

To obtain faster convergence and more accurate localization of the prediction target, EIoU loss [26] was used for bounding box regression. YOLOX-s adopts IoU loss to calculate the difference of the prediction box and the real box. However, when the prediction box and the real box do not intersect, the IoU loss [27] will exhibit gradient disappearance. The CIoU [28] considers three geometric factors, such as the overlap area, the central point distance, and the aspect ratio, to ameliorate the above problem. Due to the fact that the aspect ratio difference of CIoU cannot reflect the true discrepancy in width and height as well as their confidence, EIoU breaks down the aspect ratio factor into width and height loss separately. EIoU is defined in Equation (6). The two previous parts follow the approach in CIoU, while the width–height loss module directly minimizes the width–height difference between the target frame and the anchor frame. It effectively helps to speed up the convergence and improve positioning accuracy.

$$L_{EIoU} = L_{IoU} + L_{dis} + L_{asp} = 1 - IoU + \frac{d^2}{c^2} + \frac{\left|w - w^{gt}\right|^2}{C_w^2} + \frac{\left|h - h^{gt}\right|^2}{C_h^2} \tag{6}$$

where $L_{IoU}$ is the overlap area loss, $L_{dis}$ is the center point distance loss, $L_{asp}$ is the aspect ratio loss, $d$ represents the center distance between the groundtruth box and the prediction box, and $c$ represents the minimum border diagonal distance that can include both the prediction box and the real box, $C_w$ and $C_h$ are the width and height of the minimum bounding box covering the two boxes, respectively, $w^{gt}$ and $h^{gt}$ represent the width and height of the groundtruth box, respectively, $w$ and $h$ are the width and height of the prediction box, respectively. In summary, the illustration of EIoU loss for box regression is shown in Figure 3.

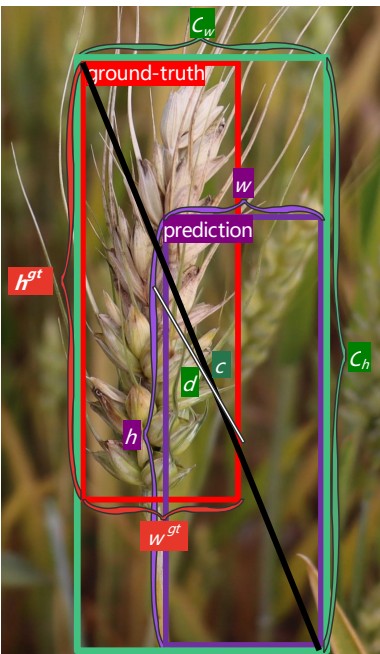

**Figure 3.** The EIoU loss for box regression. The $h^{gt}$, $w^{gt}$, $w$, $h$, $C_w$, $C_h$, $d$, and $c$ are the same definition as Equation (6).

### 2.4. Experimental Configurations and Hyperparameters Setting

The experiment employed Python as the programming language and Pytorch as the deep learning framework. The configurations of hardware and software were shown in Table 2. The hyperparameters of the GSEYOLOX-s were set as follows. The initial learning rate was 0.001, and when the loss value of the verification set did not decrease after 10 epochs, the learning rate dropped to 1/10 of the initial value. The batch size was 8, and the number of epochs (iterations) was 200. Adam optimizer was applied to optimize the model.

**Table 2.** Hardware and software configurations.

| Configuration Item | Value |
|---|---|
| CPU | Intel Xeon CPU E5-2683 v3 @ 2.00GHz |
| GPU | NVDIA Geforce RTX 3090 |
| CUDA | 11.7 |
| Memory | 64 G |
| Operating system | Ubuntu 18.04.6 LTS (64-bit) |
| Deep learning framework | Pytorch |

### 2.5. Evaluation Indices

In this study, precision, recall, F1-score, mean average precision (mAP), frames per second (FPS), and parameters are selected as evaluation indices to comprehensively evaluate the performance of deep learning modules. These evaluation indices are calculated as follows:

$$Precision = \frac{TP}{TP + FP} \times 100 \tag{7}$$

$$Recall = \frac{TP}{TP + FN} \times 100 \tag{8}$$

$$F1\text{-}score = \frac{2 \times Precision \times Recall}{Precision + Recall} \tag{9}$$

$$FPS = \frac{1}{t} \tag{10}$$

$$mAP = \frac{\sum_{i=1}^{S} AP(S)}{S} \tag{11}$$

where $TP$, $FP$, and $FN$ are the numbers of true positive cases, false positive cases, and false negative cases, respectively. Moreover, $t$ is the average time the model takes to recognize an image and $S$ is numbers of the FHB severity level. Precision is the ratio of the quantity of positive samples correctly predicted to the quantity of all positive samples predicted. Recall is the ratio of the quantity of positive samples correctly predicted to the total quantity of true samples. F1-score can be used to balance accuracy and recall. FPS is used to evaluate the real-time processing speed of a model and parameters are used to estimate the model size. The mAP is the mean of the average precision (AP) for each severity level of FHB, which is used to reflect the performance of the model. The parameters (MB) represent the spatial complexity of the model, which is calculated by the size of the convolution kernels and the number of input and output channels.

### 2.6. WeChat Mini Program Development

The GSEYOLOX-s model was deployed on the cloud server. The front-end user interface of WeChat Mini Program was written in WXML and WXSS, and its logic layer was developed in JavaScript. The application consisted of disease detection home page, encyclopedia, and detection results page. The open-source Flask framework was used to communicate between the front-end applet and the cloud server.

## 3. Results

### 3.1. Ablation Experiments

To verify the effectiveness of the G-head, SimAM, and EIoU loss function for the YOLOX-s model, ablation experiments were conducted on the FHBSGD dataset under the same experimental setting.

As shown in Table 3, the CBAM, SimAM, and EIoU all provided positive feedback to the model in terms of precision or light weight. Firstly, compared with the attention mechanism of CBAM, the SimAM achieved better results with 97.92% of mAP and without increasing the number of parameters. Secondly, although the accuracy of the model was slightly reduced by introducing G-head to YOLOX-s, the parameters of YOLOX-s + G-head were reduced by 9.84%. Additionally, when the G-head was combined with SimAM attention and EIoU loss, our proposed GSEYOLOX-s could not only reduce the parameters to 8.06 MB, but also increase the average precision of the model to 99.23%. Thirdly, as shown in Figure 4, the EIoU loss function can obtain the smaller loss value and accelerate convergence compared with the IoU loss function used in YOLOX-s. In summary, by uniformly applying the three above methods to the proposed model, the simultaneous tuning of model accuracy maximization and lightweight parameters was achieved.

**Table 3.** Ablation experiments of the GSEYOLOX-s on FHBSGD.

| Models | CBAM | SimAM | G-head | EIoU | mAP (%) | Parameters (MB) |
|---|---|---|---|---|---|---|
| YOLOX-s | | | | | 96.71 | 8.94 |
| YOLOX-s + CBAM | ✓ | | | | 97.81 | 9.03 |
| YOLOX-s + SimAM | | ✓ | | | 97.92 | 8.94 |
| YOLOX-s + G-head | | | ✓ | | 96.48 | 8.06 |
| YOLOX-s + EIoU | | | | ✓ | 98.54 | 8.94 |
| YOLOX-s + SimAM + G-head | | ✓ | ✓ | | 96.95 | 8.06 |
| YOLOX-s + SimAM + EIoU | | ✓ | | ✓ | 98.56 | 8.94 |
| **GSEYOLOX-s** | | ✓ | ✓ | ✓ | **99.23** | **8.06** |

A '✓' indicates that the corresponding module is used.

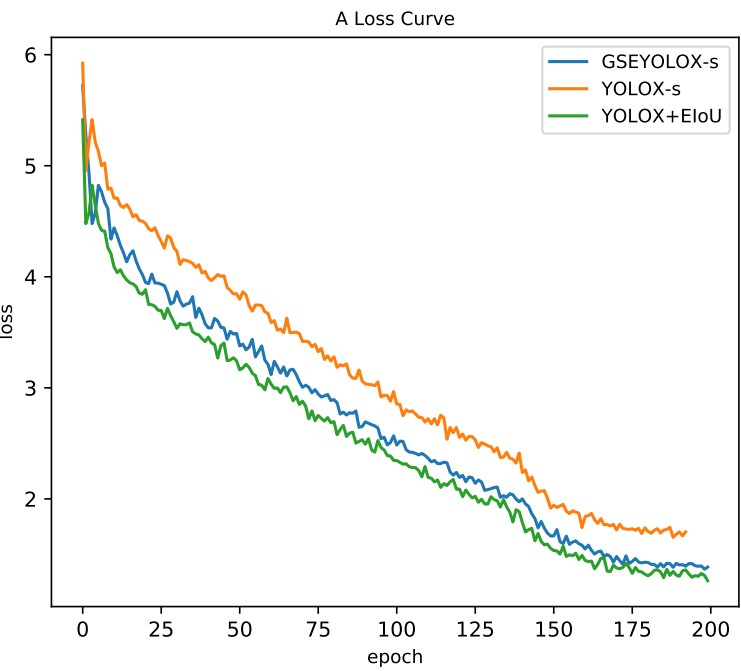

**Figure 4.** The loss curves during training process.

### 3.2. Comparison of Different Target Detection Models

The GSEYOLOX-s was compared with five typical target detection models including EfficientDet [29], Mobilenet-YOLOV4 [30], YOLOV7 [31], YOLOX-s, and YOLOX-m. The detection results on the FHBSGD are shown in Table 4. Among the five target detection models, YOLOX-s employed relatively small parameters of 8.93 MB to achieve the relatively high mAP of 96.71% and the fastest detection speed of 50 FPS, while YOLOX-m used nearly three times as many parameters (25.28 MB) to achieve the highest mAP of 98.73%, and Mobilenet-YOLOV4 achieved the highest F1-score of 94%. However, compared with the above models, the mAP, F1-score, and parameter numbers of our proposed GSEYOLOX-s were all optimized. In addition, the detection speed with 47 FPS of our model was also comparable to that of the fastest YOLOX-s, which can meet the requirements of real-time detection. Above all, the proposed model is more suitable for mobile devices.

**Table 4.** Detection results of different target detection models on FHBSGD.

| Methods | mAP(%) | F1-Score (%) | Parameters (MB) | FPS | Recall (%) | Precision (%) |
|---|---|---|---|---|---|---|
| EfficientDet | 88.71 | 86.20 | 3.87 | 11 | 81.23 | 92.21 |
| Mobilenet-YOLOV4 | 96.28 | 93.81 | 12.29 | 20 | 95.93 | 92.15 |
| YOLOV7 | 88.52 | 81.18 | 37.22 | 27 | 78.57 | 85.51 |
| YOLOX-s | 96.71 | 79.67 | 8.93 | 50 | 96.71 | 70.08 |
| YOLOX-m | 98.73 | 95.88 | 25.28 | 27 | 96.62 | 95.15 |
| **GSEYOLOX-s** | **99.23** | **96.02** | **8.06** | **47** | **96.70** | **95.33** |

In detail, the mAP of FHB severity grading detection using the six models is shown in Table 5. It can be seen that the mAP of the GSEYOLOX-s was the best among the six models and all above 99% for five FHB severity levels, while the confusion matrixes of YOLOX-s (a) and GSEYOLOX-s (b) on the test set are shown in Figure 5. Among the images of the test set, the YOLOX-s incorrectly predicted one sample for each severity level of FHB. However, the GSEYOLOX-s mis-predicted 1 Level_0 sample, 1 Level_3 sample, and 1 Level_4 sample. All samples in Level_1 and Level_2 were correctly identified. This further demonstrates the outstanding detection performance of our model.

**Table 5.** The mAP of FHB severity grading detection using six target detection models.

| Methods | mAP(%) | | | | |
|---|---|---|---|---|---|
| | Level_0 | Level_1 | Level_2 | Level_3 | Level_4 |
| EfficientDet | 70.57 | 88.56 | 91.36 | 96.28 | 96.77 |
| Mobilenet-YOLOV4 | 96.44 | 95.69 | 96.74 | 96.76 | 95.78 |
| YOLOV7 | 90.84 | 82.20 | 89.57 | 91.32 | 88.68 |
| YOLOX-s | 96.62 | 97.85 | 97.73 | 97.54 | 93.83 |
| YOLOX-m | 98.70 | 98.68 | 98.79 | 99.32 | 98.16 |
| **GSEYOLOX-s** | **99.28** | **99.23** | **99.09** | **99.33** | **99.23** |

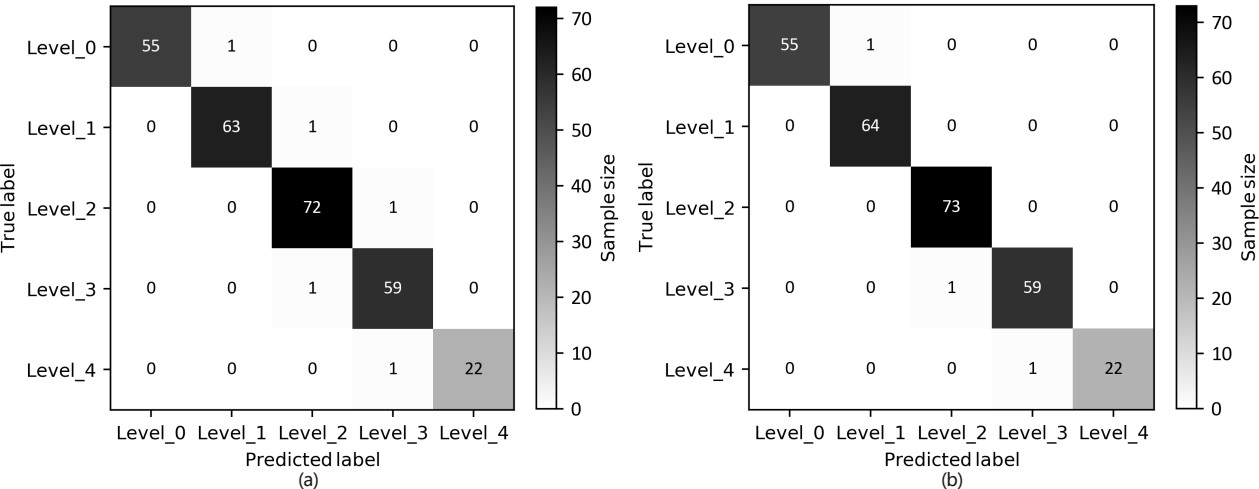

**Figure 5.** Confusion matrixes of YOLOX-s (**a**) and GSEYOLOX-s (**b**) on test set.

### 3.3. Visualization

The visual detection results of the top four precision models on FHBSGD are shown in Figure 6, and it can be seen that GSEYOLOX-s outperforms the other comparison models in terms of the accuracy of the grading identification and the prediction of the bounding box for single ear wheat images in column (a) and column (b). For the multi-ear wheat image in column 3, GSEYOLOX-s and Mobilenet-YOLOV4 can accurately detect the severity level of the two wheat ears at the same time, but for the wheat ear obscured partially in column (c), the GSEYOLOX-s detected the severity level of FHB more accurately than Mobilenet-YOLOV4. From the heatmaps in column (d), due to the application of SimAM, the GSEYOLOX-s more accurately focused on the center of the wheat than the other models, making it easier to extract the typical features of FHB. Overall, the comprehensive performance of GSEYOLOX-s was superior to other models in the identification of severity levels of FHB.

### 3.4. Model Deployment

We developed a WeChat Mini Program (Figure 7) to deploy the GSEYOLOX-s model. The service allowed croppers to upload wheat pictures taken in the field or in mobile phone albums (Figure 7a). By using our proposed model on mobile terminals, croppers can easily obtain the real-time feedback to identify the severity levels of FHB (Figure 7b). In addition, some popular science knowledge about FHB was graphically and literally provided in the platform (Figure 7c).

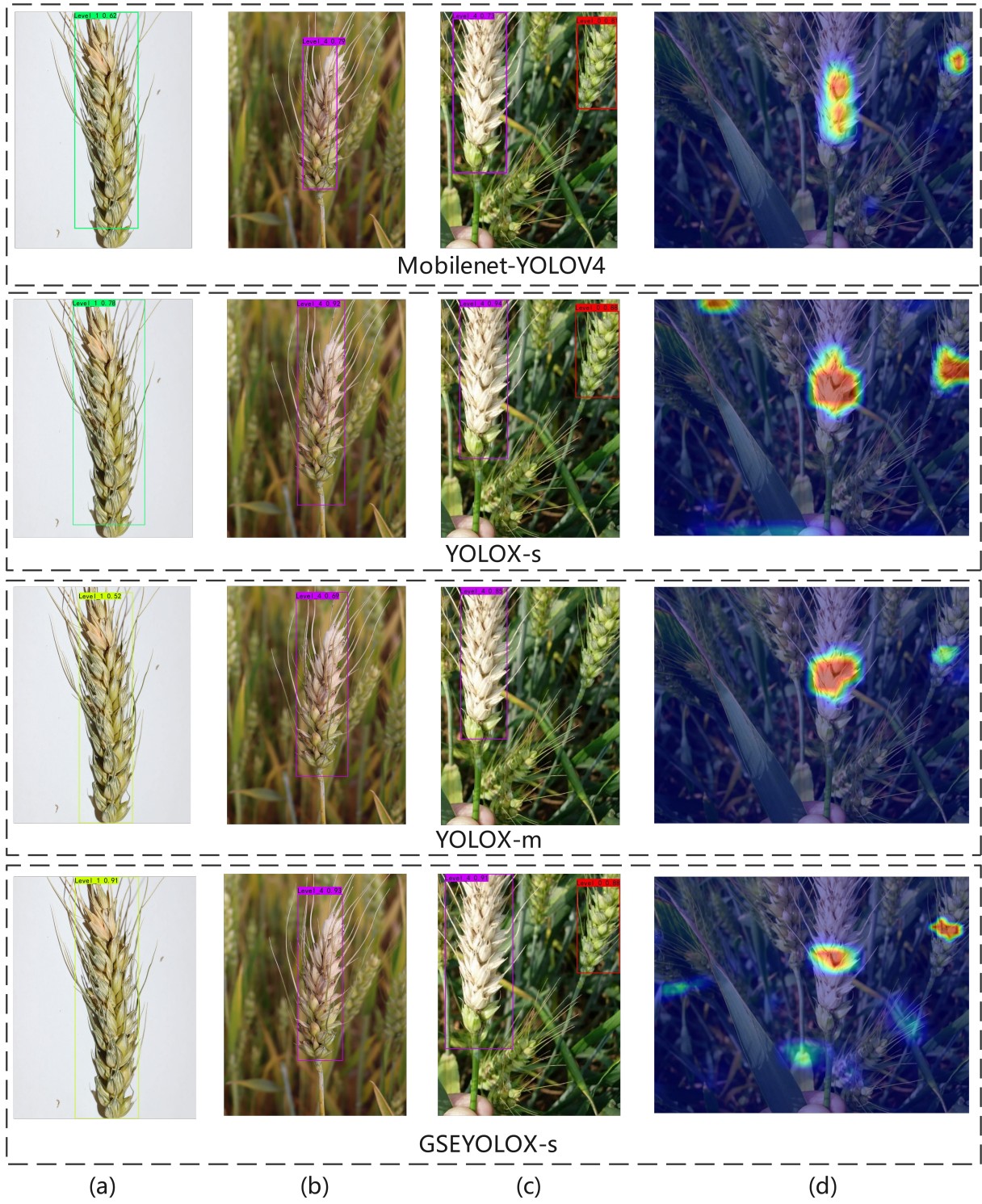

**Figure 6.** Visualization of detection results on FHBSGD. (**a**) Single wheat ear in laboratory scenes; (**b**) Single wheat ear in field scenes; (**c**) Multiple wheat ears in field scenes; (**d**) Heatmaps of detection results.

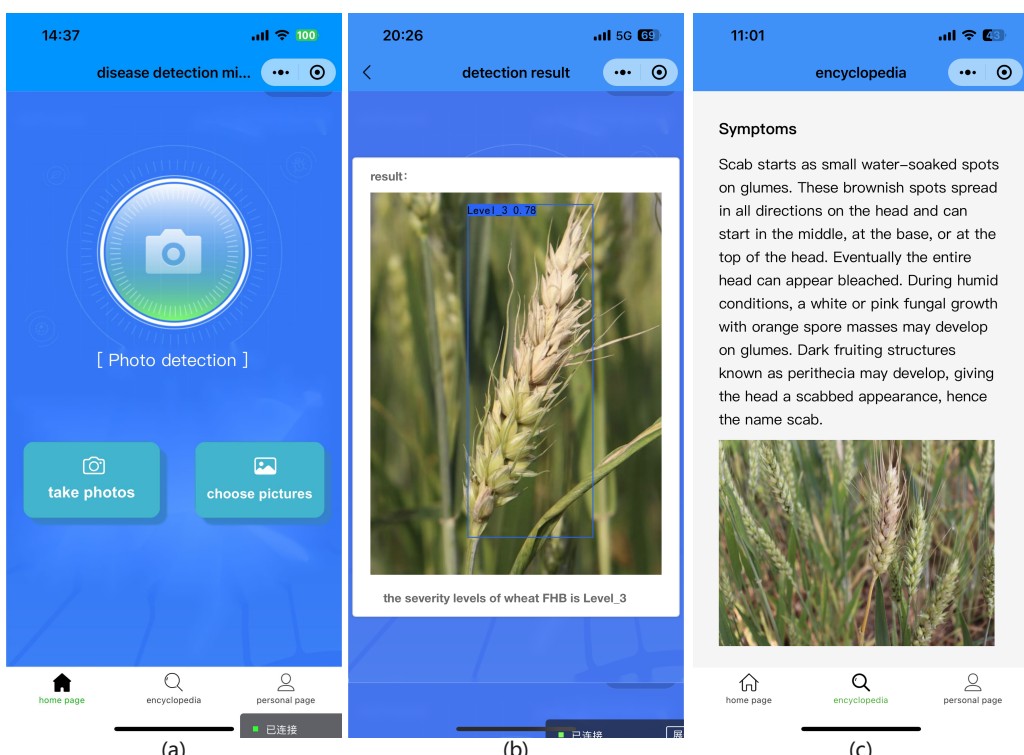

**Figure 7.** User interfaces of Wechat Mini program.

## 4. Discussion

The light weight and accuracy of the model determine whether it can be easily applied to the actual planting environment. Gao et al. [32] employed the ResNet50 model to estimate the severity of FHB. However, the model with about 25 MB parameters was constructed under simple experimental conditions, which faced the problem of decreasing accuracy in complex field conditions. The FHBSGD dataset established in this study collected disease images under experimental and complex field conditions. Training on this dataset enabled the proposed GSEYOLOX-s to learn differential features of FHB severity while avoiding the interference of complex backgrounds. Moreover, the combined optimization of SimAM, G-head, and EIoU improved the identification speed and accuracy of FHB severity, reduced the parameters of the model, and promoted the deployment of our model on the WeChat Mini Program to achieve real-time detection in the field.

FHB is a typical head disease of wheat, and directly induces the yield decline of wheat crops. Given the cost of disease control and the yield loss, Level_1 is an appropriate threshold to intervene on the disease with treatments of chemical fungicide. While the proportion of diseased spikelets is less than 25%, the intervention could retrieve a loss which far outweighs the cost of disease control. When the severity of FHB exceeds Level_1, there are currently no better control measures to save the affected crops. GSEYOLOX-s aims to accurately predict the severity of FHB as early as possible, which can help farmers in their decision toward better crop management.

## 5. Conclusions

To develop a real-time automatic identification of FHB grading, a lightweight GSEYOLOX-s model was proposed in support of the sustainable management of FHB. The SimAM attention mechanism was introduced into CSPDarkNet to enhance the ability to represent essential disease features without additional parameters. The G-head module was designed to decrease the number of parameters and ensure the performance of the model. Additionally, the use of EIoU loss further accelerated the convergence and increased the positioning accuracy of the prediction box. The results indicate that the proposed GSEYOLOX-s can

efficiently and accurately identify severity levels of the FHB with a mAP of 99.23% and 47 FPS of detection speed. Meanwhile, the proposed model was deployed on the mobile terminal to enable farmers and technicians to automatically identify the severity levels of FHB. Further work is required to enrich the construction of the canopy scale multi-severity detection model. Thus, the automatic intelligent monitoring platform of FHB will be achieved.

**Author Contributions:** R.M.: Conceptualization, Writing—Review and Editing, and Funding Acquisition; Z.W.: Writing—Original Draft, Methodology, and Investigation; F.L.: Data collation and annotation, Methodology; J.Z.: Resources and Data Verification; Y.C.: Software; X.H.: Paper Revising and Funding Acquisition. All authors have read and agreed to the published version of the manuscript.

**Funding:** This study was financially supported by the China Agriculture Research System of Wheat (CARS-03-37); Regional Innovation Capability Guidance Plan of Science and Technology Department of Shaanxi Province (2022QFY11-03); Major Science and Technology Project of Shaanxi Agricultural Collaborative Innovation and Promotion Alliance in 2022 (LMZD202203); Innovation Training Program for College Students "Identification Research for the Severity of Wheat *Fusarium* Head Blight Based on Lightweight Deep Learning Method".

**Data Availability Statement:** Data are available upon request from the authors.

**Conflicts of Interest:** The authors declare that they have no known competing financial interest or personal relationships that could have appeared to influence the work reported in this paper.

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
