# Peer review of "GSEYOLOX-s: An Improved Lightweight Network for Identifying the Severity of Wheat Fusarium Head Blight"

_agronomy, doi:10.3390/agronomy13010242_

Round 1
Reviewer 1 Report
I have identified only a few deficiencies.
P11L259: “can accurately detected“ should be “can accurately detect“.
I have no comments on the development of the object detection system. I have two comments on the phytopathological part of the paper. Firstly, disease(s) producing similar symptoms on wheat heads should be taken in consideration (used as a control) in the future. Secondly, it is too late for chemical control against FHB when symptoms appear on the heads.
Disease symptoms form later in the growing season and are not visible at spraying time. The spray window begins when most of the wheat heads on the main stems are fully emerged from the boot and continues through the time when yellow anthers form on the heads until 50% of the heads on main stems are in flower.
Weather-based FHB risk forecast maps can be used to evaluate the real-time environmental risk of FHB. Knowing the risk of temperature and rainfall to drive infection and disease development may be useful in making fungicide application decisions.
And so I foresee the main application of the model in breeding of small-grain cereals for the resistance to FHB.
I recommend accepting the paper for publication in the journal Agronomy after making minor revisions.
Author Response
Response 1:
Thanks for the comments. We re-wrote the sentence in the revised manuscript.(P10 L287)
We are grateful for the suggestion. In the future, we will conduct some early FHB prediction studies based on meteorological data. On the other hand, we will deepen the comparative studies of similar head diseases on the basis of existing research.

Reviewer 2 Report
This paper has done meaningful work towards real-time detection of fusarium head blight severity in wheat plants. The manuscript is of sufficient quality to be published, but there are some minor and major issues that need to be solved for clarification. Several comments are listed as follows:
1) Please, consider adding a graphical abstract including the developed mobile app for a better understanding of the proposed methodology.
2) In the first sentences of the Introduction section the authors could report more global information about wheat crop. In fact, the methodology proposed in the article can be easily applied all over the world and not solely in China.
3) Line 32: Authors should use the impersonal form. Please properly replace the word “us” and check throughout the manuscript.
4) Line 32: The image-acquisition procedures should be introduced before mentioning the processing of the images themselves. In this context, could be useful to introduce the type of techniques available for the image-acquisition directly in the field as well as the type of sensors commonly adopted (by highlighting the potential of handheld RGB cameras and mobile phones, which is the procedure used for the present study). Moreover, authors could briefly explain the main functioning of DCNNS in imaging-based technologies.
5) Lines 33-49: For a better comparison, authors should specify in numerical terms the number of parameters (for example the number of photos and their resolution), the storage space and the computational consumption of the previous studies reported. These data would help readers to understand what authors rightly highlighted in line 48.
6) Lines 75-93: In my opinion, all the information should be reduced in few sentences where the main objective of the article is summarily outlined. Consequently, the bulleted list should be removed and properly replaced.
7) Lines 94-97: In my opinion, authors should remove these sentences.
8) The Section 2.1 could be removed and directly replaced with Section 2.1 “Data Acquisition” and 2.2 “Data Processing”
9) The caption of Figure 1 should be more informative. In particular, authors could specify that images were acquired both in lab and field conditions.
10) Authors should specify at what stage of infection it is useful to detect fusarium disease on wheat in order to be able to intervene with treatments before irreparable damage occurs. This aspect should also be discussed in the Discussion section based on the obtained results (e.g., as reported in Figure 5) to highlight the applicability and actual usefulness of the proposed methodology in avoiding e.g., losses in grain yield and quality.
11) Line 113: Dataset was composed by images collected by 3 different sensing tools and, consequently, the spatial resolution and colorimetric information of images differed based on the relevant acquisition method. This aspect could have influenced the results. Please clarify and/or discuss.
12) If I'm not mistaken, the authors did not use a colorimetric reference during image acquisition. This aspect could have negatively influenced the accuracy of results and consequently could question the validity of the study. Please, clarify and/or discuss in detail.
13) Line 156: The full name should be reported the first time an acronym is mentioned. Please, replace with “three-dimensional (3D)”. Check throughout the manuscript.
14) Authors should be insert different sections for the Results and Discussion.
15) Section 3.1: The 201-207 lines are not appropriate for the Results section. Please, move both the sentences and Table 2 to Materials and Methods section.
16) Section 3.2: This section is not appropriate for the Results section. Please, move to a new section named “Statistical analysis” at the end of the Materials and Methods section.
17) Line 220: authors should specify, possibly in the Materials and Methods section, what the ablation experiments consist of.
18) Readers could not understand what the column Parameters (MB) of Table 3 represents. Please, specify it in the proper caption and, eventually, also in the Materials and Methods section.
19) The results shown in Table 3 are quietly similar between different models used. In this context, the contribution of the novel methodology proposed in the study could be considered not significant. For that reason, authors could add the significance (e.g., p-value) of “mAP” and “Parameters” resulting from each model with respect to the YOLOX-s values. Moreover, the number of samples analysed should be included in the caption for better completeness.
20) Lines 235-236: These sentences should be moved in the Materials and Methods section in which authors could briefly mention the main characteristic and/or differences of the models.
21) Table 4: Please, also included the values of Recall and Precision for a better comparison. Moreover, the number of samples analysed should be included in the caption for better completeness.
22) Section 3.6: The structure and operation of the developed mobile app should first be introduced in a specific section of Materials and Methods.
23) As already mentioned in my previous comment the Discussion section is missed. In particular, authors should use a dedicated Discussion section to discuss the results obtained in the study. The authors should compare the results of this work with those obtained in previous similar studies to highlight and quantify the improvement gained by the proposed methodology in the real-time detection of wheat fusarium. In addition, authors should analyse the factors that may have negatively influenced the accuracy of the results and, consequently, the improvements needed for further future applications.
24) In the Conclusion section, authors could speculate on the possible development of phenotyping pipelines for the fully automated detection (including images acquisition, images analysis and treatments distribution) of fusarium infections in wheat crops.
Author Response
We are grateful for the suggestions and comments.
Response 1: Revised as suggested, we have added a graphical abstract.
Response 2: Thank you for good points. We have revised it accordingly, please see in the revised manuscript P1 L20.
Response 3: We have corrected the inappropriate description and revised it through the whole manuscript. (P1 L33)
Response 4: We have described the DCNNS in detail in the revised manuscript, please see P1 L31-L34.
Response 5: Thank you. We have added parameters information related to storage space and computing resources in the revised manuscript, please see P1-P2, L34-L47.
Response 6: We have revised or deleted the relevant sentences, meanwhile, the bulleted lists were all removed, please see P2 L77-L86.
Response 7: Revised as suggested.
Response 8: Revised as suggested.
Response 9: We have revised the legend of Figure 1 “Wheat FHB with different severity levels under experimental and field conditions in Dataset”, please see P3 L98.
Response 10: The Level_1 is an appropriate threshold to intervene the disease with treatments of chemical fungicide. We have added the relevant descriptions in Discussion part, please see P11 L313-L320.
Response 11: Thanks for the comments. The shooting devices were a digital camera (Canon EOS 850D) and mobile phones (Apple 11pro and Huawei Mate20), using automatic white balance and optical focusing. To meet the training requirements of the proposed detection model, the image size is uniformly adjusted to the size of 640 × 640 pixels. (P3 L106-L107)
Response 12: Thanks for the comments. The dataset used for training the model is from the FHBSGD proposed in this paper. The images of the FHBSGD dataset were sampled under the guidance of planting experts. The photographic equipment used automatic white balance and optical focus. The model processing of the dataset used data enhancement method, including image cropping, scaling, shifting, blurring, blending, stitching and other processing methods, which fully considered various environmental conditions to ensure model accuracy and validity.
Response 13: Revised as suggested.
Response 14: Revised as suggested. We have stated the result and the discussion separately.
Response 15: Revised as suggested. We have moved them to Materials and Methods section. (P7 L211-L218)
Response 16: Revised as suggested. We have moved them to Materials and Methods section. (P7-P8, L219-L238)
Response 17: Thanks for the comments. In artificial intelligence, ablation experiment is a conventional method to verify the contribution of different modules to the performance of network models. It is similar to the control variable method, which controls other modules unchanged in the experiment and only change one module of the model for comparison, so as to prove the effect of the module changes.
Response 18: Revised as suggested. We have added information related to Parameters(MB) in Materials and Methods section. (P8 L236-L238)
Response 19: Thanks for the comments. Ablation experiment was conducted to verify the contribution of three improvement strategies to the performance of network models. So only GSEYOLOX-s and YOLOX-s need to be compared. Moreover, the subsequent F1-score can also illustrate the advantages of our proposed GSEYOLOX-s (96.02% vs. 79.67%). (P9, Table 4)
Response 20: Revised as suggested.
Response 21: We have added the values of Recall and Precision in Table 4. The test set information used in all experiments of this article is uniformly described in Table 1. (P9, Table 4)
Response 22: We have added the information of the developed mobile app in Materials and Methods section. (P8 L239-L244)
Response 23: Revised as suggested. We have added the discussion section. (P10 L301-L320)
Response 24: Revised as suggested. We have added the information of fully automated detection in the Conclusion part, please see P12 L333.
